# Percutaneous Electrical Nerve Stimulation (PENS) for Infrapatellar Saphenous Neuralgia Management in a Patient with *Myasthenia gravis* (MG)

**DOI:** 10.3390/ijerph20032617

**Published:** 2023-02-01

**Authors:** Sebastián Eustaquio Martín Pérez, Isidro Miguel Martín Pérez, Eleuterio A. Sánchez-Romero, María Dolores Sosa Reina, Alberto Carlos Muñoz Fernández, José Luis Alonso Pérez, Jorge Hugo Villafañe

**Affiliations:** 1Musculoskeletal Pain and Motor Control Research Group, Faculty of Health Sciences, Universidad Europea de Canarias, 38300 Santa Cruz de Tenerife, Spain; 2Musculoskeletal Pain and Motor Control Research Group, Faculty of Sport Sciences, Universidad Europea de Madrid, 28670 Villaviciosa de Odón, Spain; 3Departamento de Farmacología y Medicina Física, Área de Radiología y Medicina Física, Secciones de Enfermería y Fisioterapia, Facultad de Ciencias de la Salud, Universidad de La Laguna, 38200 Santa Cruz de Tenerife, Spain; 4Escuela de Doctorado y Estudios de Posgrado, Universidad de La Laguna, 38200 Santa Cruz de Tenerife, Spain; 5Department of Physiotherapy, Faculty of Sport Sciences, Universidad Europea de Madrid, 28670 Villaviciosa de Odón, Spain; 6Physiotherapy and Orofacial Pain Working Group, Sociedad Española de Disfunción Craneomandibular y Dolor Orofacial (SEDCYDO), 28009 Madrid, Spain; 7Onelife Center, Multidisciplinary Pain Treatment Center, 28925 Alcorcón, Spain; 8IRCCS Fondazione Don Carlo Gnocchi, 20148 Milan, Italy

**Keywords:** *Myasthenia gravis*, neuralgia, saphenous nerve, PENS

## Abstract

*Myasthenia gravis* is a neuromuscular transmission disorder characterized by weakness of the cranial and skeletal muscles, however, neuropathies are extremely rare. In this case report we present a case of a 61-year-old man diagnosed *Myasthenia gravis* who came to our attention due to a 1 week of acute deep pain [NPRS 8/10] in the anterior and medial right knee which occurred during walking [NPRS 8/10] or stair climbing [NPRS 9/10]. A complete medical record and clinical examination based on physical exploration and ultrasound assessment confirmed a infrapatellar saphenous neuralgia. Therapeutic interventions included Percutaneous nerve electrical stimulation combined with pain neuroscience education, neural mobilization of the saphenous nerve and quadriceps resistance exercises. After 4 weeks, pain intensity [NRPS = 1/10], knee functionality [OKS = 41/48] and lower limb functionality [LLFI = 80%] were notably improved, nevertheless, fatigue [RPE = 2/10] was similar than baseline. At 2 months of follow-up, the effect on intensity of pain NRPS [0/10] and functionality OKS [40/48] and LLFI [82%] was maintained, however, no significant clinical changes were detected on perceived fatigue RPE Scale [2/10]. Despite the important methodological limitations of this study, our case report highlights the efficacy of percutaneous electrical nerve stimulation combined with physical agents modalities for pain and functionality of infrapatellar saphenous neuralgia in the context of *Myasthenia gravis.*

## 1. Introduction

*Myasthenia gravis (MG)* is a disorder of neuromuscular transmission characterized by weakness of cranial and skeletal muscles [1]. Autoantibodies directed against acetylcholine receptors damage the motor endplate portion of the neuromuscular junction [2]. The impairment of the transmission of impulses to skeletal muscles causes muscle weakness and fatigue. Muscle weakness affects the cranial muscles compared to the muscles of the limbs and is associated with movement [3]. Often, the outer muscles of the eye are affected, with ptosis and diplopia. Since proximal muscles are more affected than distal muscles, the bulbar muscles are also affected, resulting in dysarthria and nasal murmurs [4,5]. In addition, tendon reflexes were normal and sensitivity was normal. While MG is generally not a painful condition, some patients develop certain orofacial, spinal or even limb pain syndromes that may be more common than healthy individuals.

In treating these symptoms, we have identified pharmacological approaches (anticholinesterase drugs, corticosteroids, immunosuppressants and others such as plasma exchange and immunoglobulins) and non-pharmacological approaches based on the application of electrotherapy to relieve pain and restore limb function learning method [6,7,8]. One of the methods of greatest interest in the physical therapy community is the use of percutaneous electrical nerve stimulation (PENS), a minimally invasive electrical therapy technique designed to stimulate one or more individual nerves or dermatomes through the insertion of a needle usually guided by ultrasound imaging [9]. This technique involves inserting a needle into the nerve or soft tissue surrounding the dermatome it innervates, while attaching a grounding pad to the skin near the target tissue. The pen is connected to a low-voltage pulse generator that can provide a low-frequency background current. Depending on the applied voltage threshold, we can activate different types of sensory fibers or motor neurons according to the defined therapeutic target. A session can usually last 15 to 60 min.

Current literature suggests that the technique is effective, and safe, with no serious side effects other than a slight increase in pain and petechiae due to minor bleeding during needle manipulation. In terms of its clinical efficacy, randomized controlled trials published to date suggest that PENS is effective in the treatment of certain clinical conditions associated with primary neuropathic pain, such as cervical or low back pain, or secondary diabetes or osteoarthritis and even postoperative pain [10,11,12,13,14,15,16,17].

Despite clinical efficacy in these disorders, there are no studies analyzing the role of PENS in the management of pain and functional knee dysfunction in patients with MG [18]. Therefore, this case report aims to highlight the use of a promising electrical analgesia technique for the management of pain associated with MG. This is a novel application not previously described in the literature and is a starting point to investigate whether improving pain management in patients with this neuromuscular junction (NMJ) disorder has potential clinical benefits.

## 2. Objectives

The main objective of this work was to assess the effectiveness of PENS within a multimodal physical agent treatment for infrapatellar saphenous neuralgia management in a patient with *Myasthenia gravis* (MG).

## 3. Case Description

This case report was written following the CARE guidelines, and the authors received written informed clinical consent from the patient [19]. A 61-year-old male with a diagnosis of MG 12 years ago with no previous history presented acute and deep pain (NPRS 8/10) in the anterior and medial aspect of the right knee for 1 week until the time of consultation. It manifested itself during prolonged sitting (NPRS 7/10) and during standing activities such as walking (NPRS 8/10) or going downstairs (NPRS 9/10). He also reported changes in sensitivity in the anterior area of the knee, as well as recurring bouts of bilateral strength loss in the legs and generalized fatigue in the quadriceps area. Figure 1 shows a body chart and knee pain map of the right knee.

The onset of symptoms was described by the patient as abrupt without being associated with previous trauma. In addition, he pointed out that when he consumed analgesics, such as ibuprofen, they did not completely relieve him of pain, with it being moderately disabling. Regarding medical history, the patient had worked as an office worker prior to illness and admitted to being sedentary or inactive IPAQ (<5000 steps/day). Additionally, he stated that he is a smoker (1 pack per day) and an occasional alcohol drinker. He also noted that he had been suffering from benign prostatic hyperplasia for 10 years and was medicated with anticholinergic drugs.

## 4. Clinical Examination

An anamnesis was carried out identifying affiliation characteristics, as well as possible antecedents within his MG diagnosis, such as previous illnesses or taking medication, and finally clinical aspects related to pain and other possible factors triggering the symptoms. The results of this procedure are summarized in the following Table 1.

The patient had no family history of MG and was diagnosed 12 years ago and treated with the anticholinergic Mestinon (60 mg) in 3 tablets per day, the cumulative daily dose being a total of 180 mg. In addition, the patient was regularly treated with antiretroviral therapy for acquired immunodeficiency syndrome (AIDS). Regarding to the course of MG it was variable as there were at least 3 exacerbations of lower extremity fatigue symptoms in the past three years.

A comprehensive assessment was carried out, starting with the detection of red flags, which will allow us to screen patients for medical referral if necessary. Once it was determined that his symptoms of pain, dysesthesia and weakness were not due to primary or secondary motor neuron disorder compatible with lumbosacral plexus inflammation, irritation or mechanical compression, it was decided to continue with local exploration to reproduce their symptoms.

The physical examination began with an examination of the right knee and aimed to detect changes in the appearance of the tibiofemoral and patellofemoral joints compatible with inflammation (*tumor, redness, sweating*, etc.). This procedure was complemented with superficial palpation aimed at identifying the existence of changes in temperature. Superficial sensitivity was assessed through a von Frey monofilament that showed a slight decrease (0.1 Kg/Pa) in comparison to contralateral side (0.07 Kg/Pa). Furthermore, a decrease in the pain pressure threshold (PPT) of the anteromedial surface of the tibiae was evaluated through a digital Pain Tester (*FPX TM Algometer*, *Wagner*), showing for the painful side (1.22 Kg/Pa) versus left side (2.17 Kg/Pa). As no significant abnormalities of a clear inflammatory process were revealed, suggestive of infrapatellar plica impingement or bursitis, it was decided to continue with the osteoarticular and neuromuscular exploration of the joint.

First, an assessment of active physiological movements of the knee in flexion and extension was performed with different degrees of axial load in sitting position, and bipodal and monopodal load on the affected knee, detecting an increase in symptoms of pain, dysesthesia and perceived fatigue when the patient performed several repetitions of a single leg squat. Accessory mobilizations in the postero-anterior direction and medial and lateral rotations in the tibio-femoral joint and lateral glide, medial and lateral rotations and ascents and descents of the patellofemoral joint were also carried out with the aim of identifying tissues capable of increasing symptoms, but no significant changes were detected.

Secondly, we carried out a neuromuscular examination aimed at identifying the existence of myofascial pain through palpation of the *vastus medialis* (VM) muscle (MTrP1, MTrP2) and quadriceps *rectus femoris* (RA) (MTrP1), identifying only increased symptoms of pain (NPRS 9/10) and dysesthesia on palpation of the *vastus medialis* (MTrP1). Quadriceps muscle strength was also evaluated using a BioFET Dynamometer (*Mustec, Muscle Dynamic Technology b.v*.), finding a slight decrease in maximal voluntary isometric contraction of the quadriceps with respect to the contralateral limb measured in three repetitions with a rest period between evaluations of 1 min (20.98 Kg, SD 1.2 vs. 22.34 Kg, SD 1.0). Finally, the patellar osteotendinous reflex was evaluated without any findings of hyporeflexia or areflexia compared to the contralateral.

Applying a differential diagnosis strategy, we decided to continue the neural examination of the infrapatellar branches of the saphenous nerve (IBSN) which, in the case of neuropathy, produces symptoms in an area reasonably similar to that of the VM myofascial trigger point. To assess this, we decided to perform an orthopedic neural mechanosensitivity maneuver (*Prone Knee Bending Test*) that could provoke an increase in pain intensity (NPRS 9/10) and diffuse dysesthesia recognizable by the patient in the medial zone of the knee.

Upon detecting it, we decided to carry out an ultrasonography assessment of the IBSN, by first performing an axial compression on the nerve path with the ultrasound transducer to induce pain, somatosensorial signs (i.e., *paresthesias*) or motor output disturbances (i.e., *loss of strength*) described previously in the literature for other nerves [21,22]. On examination, we were able to elicit all the symptoms reported by the patient, although no sonographic changes suggestive of a nerve space conflict were detected. This allowed us to conclude the existence of IBSN sensory nerve neuralgia.

To complete the exploration, we carried out a baseline measurement of intensity of pain, knee and lower limb functionality and perceived fatigue that would serve to objectify changes throughout the treatment follow-up.

## 5. Intervention

Once the diagnosis was reached, it was decided to implement a treatment strategy aimed at reducing neuropathic pain and improving sensitivity in the anteromedial region of the knee. It was decided to carry out a multimodal intervention based on PENS combined with PNE, neural mobilization of saphenous nerve and quadriceps resistance exercise.

### Ultrasound Nerve Identification

The identification of the nerve by ultrasound was carried out by a physical therapist with 10 years of experience following the method described by Riegler et al. (2018) using B-mode ultrasound imaging (4.2 to 13 MHz) coupled with a multifrequency linear array transducer, model 12L-RS (*GE Logiq V2, GE Health Care, Chicago, IL, USA*) [23]. Ultrasonography scanning began with the patient in supine position with knees straight placing the transducer in the horizontal plane to obtain a cross-sectional view of the m. *sartorius* (ST) and m. *vastus medialis* (VM) approximately 10 to 15 cm above the knee joint line. ST was shielded distally until reaching a tubular structure that pierces m. *tensor fasciae latae* (TFL) over the ST, and the nerve located in the fat pad is IPBSN. With the purpose of correctly assigning the IPBSN to the origin of the saphenous nerve, the former was traced proximally to its origin. The IPBSN is then traced distally, across the medial tibiofemoral joint line, until the branch divides into terminal branches. When detectable, all branches are followed until reaching their most distally visible part. Figure 2 shows the *ultrasound nerve identification.*

PENS was administered using the ES-130 stimulator (*Ito Co., Ltd., Osaka, Japan*) and Agu-Punt APS needles (*Agu-Punt S.L., Barcelona, Spain*) with dimensions of 0.25 × 30 mm, making the entry of the needle using a short-axis plane, locating the needle in the region of the perineurium of bifurcation of the infrapatellar branch of the saphenous nerve (IPBSN) before its division into superior and inferior branches. To confirm the location of the needle, the PENS was tested on several occasions, verifying that the stimulated area coincided with the dermatomal pattern reported by the patient. Stimulation was maintained between 200 Hz and 250 Hz at an intensity of 0.8 to 1.0 mA depending on patient tolerability for 45 min at the first consultation. Figure 3 shows the *percutaneous electrical nerve stimulation of the saphenous nerve.*

Only in the first session, a face-to-face and individual Pain Neuroscience Education (PNE) session was carried out, which served to reconceptualize neuropathic pain through two metaphors: (1) *Alarm system*: your nerves working as a system of alarm to protect you; and (2) *Nerve sensors*: nerve sensors that tell you about movement, stress and cold [24]. In the following sessions and until the end of the treatment, the neural mobilization of the saphenous nerve was implemented with the patient in lateral decubitus with the hip extended and abducted and the knee flexed. The physiotherapist passively applied 10 series of 10 reps towards hip extension with a frequency of 2 Hz up to a maximum of 10 min of exposure [25]. In addition, quadriceps resistance exercises were also incorporated from the second session following the following sequence: *Supine bridge, Supine single bridge, Squat* and *Single Squat*, with a maximum dose of 3 series, 10 reps, 30 s rest/series in the time of discharge. The intervention and distribution of the PENS intervention applied to the patient is explained in Table 2.

## 6. Follow-Up

The follow-up was carried out during the 2 months after discharge, and in which time, the patient was contacted by telephone, and the same instruments used during treatment were sent to him by email to proceed with the completion. First, the patient stated that the pain had completely disappeared and that he had not suffered any new relapse during this time. He noted the occasional existence of hypersensitivity in the area that decreased with the application of cryotherapy in the area. In relation to knee function, a slight improvement was maintained in line with those obtained during treatment. In this section, the patient obtained an OKS score of 40/48, with loaded movement activities being those that he continued to perceive as problematic. In addition, the global functionality of the lower limb (LLFI) remained at similar values from the date of discharge at 82%. Finally, the level of fatigue during the basic activities of daily living persisted in figures like those obtained during the RPE treatment (2/10). Lastly, no relevant adverse events were detected during follow-up.

## 7. Discussion

This case report describes the history, clinical examination and management of anteromedial knee pain caused by IBSN neuralgia in a patient with MG.

Knee saphenous neuralgia is the result of nerve compression mediated by posttraumatic fibrosis or neuroma [26]. Neuralgia has been reported as a concomitant injury to ACL surgery, knee injections and other procedures. Neurolysis and nerve transplantation are the most used methods for the treatment of saphenous neuralgia [27,28]. However, all surgical procedures carry risks due to possible complications or simply for cosmetic reasons [29].

In addition to knee saphenous neuralgia, the patient also suffered from MG. Pain in this condition is rare. In the context of MG, pain can be explained by several circumstances as physical causes, such as weakness or postural changes in patients due to a sedentary lifestyle, or psychiatric disorders such as mood or sleep disturbances [30].

Managing neuropathic pain in patients with MG can be challenging. Painkillers can even alter the effects of medications used to treat MG [31,32]. For example, antidepressants or anticonvulsants, often used to treat migraines, can make MG symptoms worse [33].

When the disease affects the respiratory muscles, taking opioids such as morphine or oxycodone can be dangerous, even fatal, because they can significantly reduce breathing rate [34,35]. Any alternative must not only be effective, but also safe and have fewer side effects than drug or surgical options. Any alternative must not only be effective, but also safe, and cheap, and have fewer side effects than pharmacological or surgical options. As shown in this case report, PENS together with PNE, neural mobilization and resistance exercise could be a good alternative for the treatment of IBSN neuropathic pain in the context of a patient diagnosed with MG.

However, a set of limitations has been detected that affect the external validity of the results. First, from the methodological point of view, the study design is only valid to identify the potential of PENS in the case presented and does not allow us, under any circumstances, to extrapolate the results to a larger population.

Secondly, the limited availability of clinical practice guidelines for the management of neuropathies in neuromuscular diseases of the motor endplate may contradict the rationale that we have followed when selecting and ordering the intervention in this patient.

Third, the absence of randomized clinical trials that guarantee the standardization of the procedure used for this patient introduces uncertainty as to whether the dose of PENS used would be adequate for IBSN neuropathic pain.

Fourthly, the PENS intervention combined with other treatments such as hands-off (PNE and resistance exercise) and hands-on methods (neural mobilization of saphenous nerve) may have an “*overdose*” effect that affects the measurement of the variables and outcomes that have been studied in this case report. In line with this, the release of substances at the peripheral, medullary, and central level for descending control of pain could have been elicited with any of the treatments that have been proposed in the intervention plan, with which the overall effect could not be exclusively attributed to PENS.

## 8. Patient’s Point of View

The patient stated that he was satisfied with the use of a new minimally invasive treatment that improved his pain and improved the functionality of his knee. He reports that his quality of life has improved, and despite his MG disease, he is able to perform most activities of daily living. To objectify this change, a Global Rating of Change questionnaire (GROC) was administered to assess the change perceived by the patient, indicating a 6/7 or much better change that he had noticed at the end of the treatment.

## 9. Conclusions

In conclusion, the combination of PENS with other physical agents modalities has been shown to be effective in the management of pain and function in IBSN neuralgia in patients with MG. However, the lack of standardization in diagnostic and intervention procedures requires randomized clinical trials to determine the effective dose and appropriate combinations with other treatment modalities. In addition, we must be mindful of the need for proper training of physical therapists in screening skills for neurological injury, as well as specific skills in using ultrasound as a guide for minimally invasive treatments.

## Figures and Tables

**Figure 1 ijerph-20-02617-f001:**
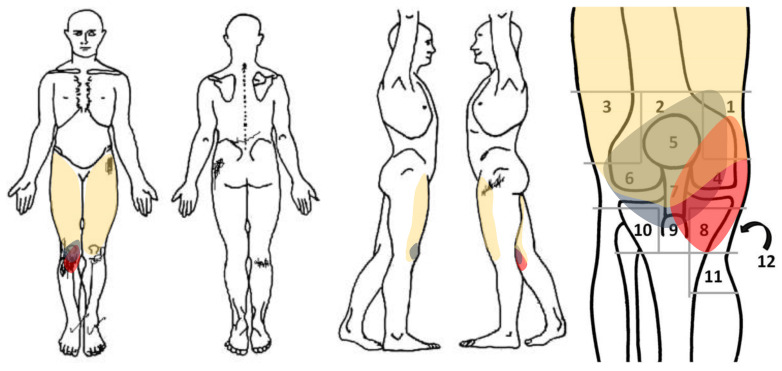
Body chart and knee pain map of right knee. Note: (1) anterior supero-medial (m. *vastus medialis*), (2) anterior superior (*quadriceps tendon*), (3) anterior supero-lateral (*iliotibial band* and m. *vastus lateralis*), (4) anterior medial (medial joint line and medial collateral ligament), (5) patella, (6) anterior lateral (lateral joint line and lateral collateral ligament), (7) patella tendon, (8) pes anserinus, (9) tibial tuberosity, (10) anterior infero-lateral (fibula head and tibialis anterior), (11) anterior infero-medial (tibial insertion of the superficial medial collateral ligament), (12) popliteal fossa. The red circle indicates pain, the grey circle indicates dysesthesia, and the yellow circle means fatigue. Adapted from Ikeuchi et al. *Springerplus*, 2013 [20].

**Figure 2 ijerph-20-02617-f002:**
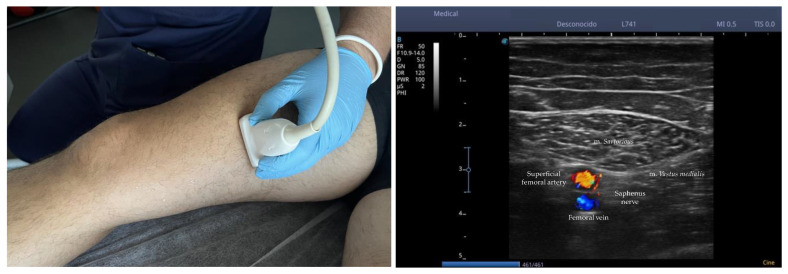
Ultrasound nerve identification. Note: During ultrasound identification of the saphenous nerve, the probe is positioned transversely on the medial aspect of the thigh following the course of the nerve distally. Before its division, the nerve is located at a crossroads comprised of the m. *sartorius* on the superficial plane, the m. *vastus medialis* on the medial side and surrounded by the femoral vein (*blue*) and the superficial femoral artery (*red*).

**Figure 3 ijerph-20-02617-f003:**
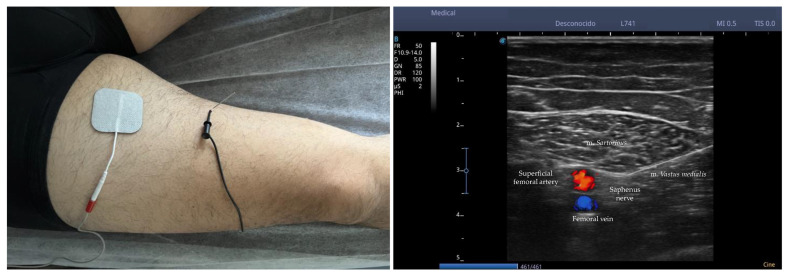
Percutaneous electrical nerve stimulation of saphenous nerve. Note: Once the nerve was identified, the needle was inserted parallel to the plane of the probe between the muscular septum of the m. *sartorius* and the m. *vastus medialis* locating the tip of the needle in the perineurium of the saphenous nerve.

**Table 1 ijerph-20-02617-t001:** Description of clinical characteristics of subject.

Affiliation Characteristics	
Age, yrs	61.0
Gender	Male
Job	Office worker
**Anthropometric Measures**	
Weight, Kg	89.1
Height, cm	178.0
BMI, Kg/cm^2^	28.1
**Disease Status**	
Age at onset of MG, yrs	49.0
Duration of MG, yrs	12.0
Drugs intake	Anticholinergics (Mestinon, 60 mg)

**Table 2 ijerph-20-02617-t002:** Intervention and clinical effects measured by instrument.

Baseline	Intervention	Clinical Effect
	**Anamnesis** **Clinical examination** **Pain neuroscience education** *- Content:* (1) *Alarm system:* Your nerves working like an alarm system to protect you (2) *Nerve sensors:* Nerve sensors telling you about movement, stress and cold - *Time of exposure:* 15 min **PENS of sensitive IPBSN** - *Dose:* 250 Hz 1.0 mA - *Time of exposure:* 45 min	- Pain (NRPS) = 9/10 - Knee functionality (OKS) = 16/48 - Lower limb functionality (LLFI) = 15% - Fatigue (RPE Scale) = 5/10
**First session** **(Post-1 week)**		
	**PENS of sensitive IPBSN** *- Dose:* 250 Hz 1.0 mA - *Time of exposure:* 30 min **Neural mobilization of IPBSN** *- Dose:* 10 series, 10 reps 2 Hz - *Time of exposure:* 10 min	- Pain (NRPS) = 8/10 - Knee functionality (OKS) = 17/48 - Lower limb functionality (LLFI) = 25% - Fatigue (RPE Scale) = 3/10
**Second session** **(Post-2 week)**		
	**PENS of sensitive IPBSN** *- Dose:* 250 Hz 1.0 mA - *Time of exposure:* 25 min **Neural mobilization of IPBSN** *- Dose:* 10 series, 10 reps 2 Hz - *Time of exposure:* 10 min **Quadriceps resistance exercise:** - *Dose*: (1) Supine bridge 3 series, 10 reps, 10 seg contraction, 10 seg rest/series (2) Supine single bridge 2 series, 10 reps, 10 seg contraction, 10 seg rest/series - *Time of exposure:* 25 min	- Pain (NRPS) = 6/10 - Knee functionality (OKS) = 20/48 - Lower limb functionality (LLFI) = 40% - Fatigue (RPE Scale) = 3/10
**Third session** **(Post-3 week)**		
	**PENS of sensitive IPBSN** *- Dose:* 250 Hz 1.0 mA - *Time of exposure:* 25 min **Neural mobilization of IPBSN** *- Dose:* 10 series, 10 reps 2 Hz - *Time of exposure:* 10 min **Quadriceps resistance exercise:** - *Dose*: *(1) Supine single bridge* 3 series, 10 reps, 10 seg contraction, 10 seg rest/series *(2) Squat* 2 series, 10 reps, 30 seg rest/series - *Time of exposure:* 25 min	- Pain (NRPS) = 2/10 - Knee functionality (OKS) = 33/48 - Lower limb functionality (LLFI) = 70% - Fatigue (RPE Scale) = 3/10
**Discharge** **(Post-4 week)**		
	**PENS of sensitive IPBSN** *- Dose:* 250 Hz 1.0 mA - *Time of exposure:* 25 min **Neural mobilization of IPBSN** *- Dose:* 10 series, 10 reps 2 Hz - *Time of exposure:* 10 min **Quadriceps resistance exercise:** - *Dose*: *(1) Squat* 3 series, 10 reps, 30 seg rest/series *(2) Single Squat* 2 series, 10 reps, 30 seg rest/series - *Time of exposure:* 25 min	- Pain (NRPS) = 1/10 - Knee function (OKS) = 41/48 - Lower limb functionality (LLFI) = 80% - Fatigue (RPE Scale) = 2/10

## Data Availability

Data will be available on demand.

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
