# Peer review of "Percutaneous Electrical Nerve Stimulation (PENS) for Infrapatellar Saphenous Neuralgia Management in a Patient with Myasthenia gravis (MG)"

_ijerph, 2023, doi:10.3390/ijerph20032617_

Round 1

Reviewer 1 Report

Dear Authors:

First of all I would like to congratulate you for carrying out this research work on a rare topic in the healthcare field and more so in physiotherapy. Next, I would like to suggest certain aspects for improvement:

- In the introduction section you should add the objective of your study.

- In the clinical examination section, in the table, it does not provide us with much information if the patient is married or not, in my opinion, this line should be removed. In addition, they could refer to the time that the patient has had the disease and not the patient's years when the disease debuted. In the same table, they should remove the male/female gender, as it is already stated next to it. 

- In the intervention section, Figure 2 shows how the puncture is carried out by means of an ultrasound study, and later in Figure 3 the needle is illustrated by an arrow. It is possible to improve the ultrasound image with the real image of the needle inside the patient's body together with the ultrasound image and adding echo-Doppler to check that the needle does not reach the vein or artery, since it is very close, as shown in Figure 2. In addition, they refer to the fact that they perform saphenous nerve mobilization, but do not explain how it is performed, with what technique, or for what purpose. It would be good to add a photo or sequence of this maneuver.

- Regarding the follow-up section, they carry out a review of the patient by means of a telephone interview. This method represents a very important bias for the research. They should justify this section or look for a much more reliable way of comparison.

- In the discussion section, they should include the limitations of their study.

Best regards.

Author Response

Dear reviewer,

Thank you for your suggestions.

Attached we send you a cover letter with changes required.

Thank you,

Kind regard

Reviewer 2 Report

Thank you for the opportunity to review this manuscript entitled “Percutaneous electrical nerve stimulation (PENS) for an infrapatellar saphenous neuralgia management in a patient with myasthenia gravis (MG) ”. This is a new a new electrotherapy technique for new approach and management of pain and dysfunctions.

The authors have conducted a case study of neuropathic neuralgia of sensitive nerve over the anterior and medial aspect of the right knee.

I think this is a good case to improve our knowledge and tools to treat the above-mentioned cases, and It could encourage other clinicians to use this technique and researchers to lead new research.

Overall, I found some aspects should be reviewed.

Abstract: it is no clear how you present your case. First of all, you introduce the Myasthenia Gravis but then you speak about neuralgia of the anteromedial knee joint.

Review from line 16 to 18 of the abstract.

Line 42: “Often, the outer muscles of the eye are often affected “; often is reiterative.

Line 56: “inserting a needle or needle into the nerve or soft tissue”; Rewrite this sentence.

Line 80-81: “[NPRS 7/10] and during standing activities such as walking NPRS [8/10] or going downstairs [NPRS 9/10]”; you should use the same style NPRS into the brackets or out. 

Line 83-84: this is probably a mistake.

Figure 1: review the zones are illustrated. It has an error.

Line 101: It isn’t needed to refer the title of the table in the text.

Line 145: “test (Prone Knee Bending Knee Bending Test)”; are you speaking of the Prone Knee Bending Test? Correct it.

Line 149-150: “to detect a possible entrapment of this sensory branch that has been described in the literature.”; you should the provide the reference what confirm this asseveration.

Line 161: You have to inform in right way of the ultrasound device; e.g.: We used B-mode ultrasound imaging (4.2 to 13 MHz) coupled with a multifrequency linear array transducer, model 12L-RS (GE Logiq V2, GE Health Care, Chicago, IL, USA).

Line 166: “examined distally until a tubular structure is reached that pierces the m. tensor fasciae latae (TFL)”. Are you sure distally you can find the TFL close to ST? 

Line 255: the first time of acronym appear it should be defined; Global Rating of Change (GROC). 

Author Response

Dear reviewer,

Thank you for your suggestions.

Attached we send you a cover letter with the changes required.

Thank you,

Kind regard
